# Determinants of Pet-Friendly Tourism Behavior: An Empirical Analysis from Chile

**DOI:** 10.3390/ani15121741

**Published:** 2025-06-12

**Authors:** Iván Veas-González, Manuel Escobar-Farfán, Nelson Carrión-Bósquez, Jorge Bernal-Peralta, Elizabeth Emperatriz García-Salirrosas, Sofía Romero-Contreras, Camila Díaz-Díaz

**Affiliations:** 1Departamento de Administración, Facultad de Economía y Administración, Universidad Católica del Norte, Antofagasta 1270709, Chile; iveas@ucn.cl (I.V.-G.); nelson.carrion@ucn.cl (N.C.-B.); sofia.romero@alumnos.ucn.cl (S.R.-C.); camila.diaz01@alumnos.ucn.cl (C.D.-D.); 2Department of Administration, Faculty of Administration and Economics, University of Santiago of Chile (USACH), Santiago 9170020, Chile; 3Facultad de Administración y Economía, Universidad de Tarapacá, Arica 1000007, Chile; 4Faculty of Management Science, Universidad Autónoma del Perú, Lima 15842, Peru; egarciasa@autonoma.edu.pe

**Keywords:** pet-friendly tourism, travel intention, pets, attitude

## Abstract

Pet tourism is growing as more people consider their pets family members and want to include them in their travels. Our study examined what motivates Chilean pet owners to travel with their pets by looking at emotional connection, perceived benefits, social status, and new experiences. We found that people are more likely to travel with their pets when they see clear benefits and new experiences rather than due to emotional attachment alone. These findings help tourism businesses better serve pet owners by focusing on practical benefits and unique experiences, making it easier for families to travel with their pets.

## 1. Introduction

Nowadays, domesticated animals or so-called pets are considered family members, playing a fundamental role in the emotional well-being of their owners [1,2,3]. This trend is reflected in global statistics: approximately 65% of American households own at least one pet [4], while in the United Kingdom, this percentage reaches 51% of households [5]. In Latin America, the pet industry has experienced significant growth. Brazil led the regional market by generating more than USD 7.9 billion in revenue during 2022, followed by Mexico and Argentina with USD 3.3 and USD 1.2 billion, respectively. In Chile, where 84% of households have at least one pet, the pet food and product market reached USD 1 billion in 2022 [6]. These figures demonstrate that domestic animals not only have a significant presence in daily family life, to the point of being included in traditional human activities such as vacations and tourist visits, but also represent a constantly growing economic sector [7,8,9].

In this sense, pet tourism has experienced significant growth in recent years, with an increase in the number of people who consider their pets as family members and want to include them in their travel experiences [10,11]. Pet tourism refers to travel activities in which pet owners are accompanied by their pets, integrating the pets’ presence into the overall tourism experience. This concept involves not only the logistical aspects of traveling with animals but also the emotional, social, and psychological motivations that drive owners to include their pets in leisure travel [10,11]. Pet owners are increasingly seeking destinations and services that are friendly to their animal companions, leading to the development of a specialized segment in the tourism industry [9,12]. This phenomenon has generated a significant economic impact, with owners willing to pay more for services that accommodate their pets [7]. Although traveling with pets presents limitations that may inhibit tourism participation [13], in Chile, the percentage of people traveling with their pets increased from 14% to 17% between 2019 and 2022 [14]. In response to this growing trend, companies such as Uber have adapted their services, launching “Uber Pet” in 2022. This modality allows traveling safely with pets through specific policies and an additional cost, thus serving a market where 84% of Chilean households have at least one pet [15]. Additional examples of pet-friendly services and destinations include restaurants, airlines, hotels, and resort centers [14].

Various factors, including emotional attachment, influence the decision to travel with pets [16], perceived benefits [13], social prestige [17], and the search for novel experiences [18]. Although pet tourism has received growing academic attention in recent years—particularly in North American and Asian contexts—there is a lack of empirical research focusing on Latin America. This regional gap is significant, considering the increasing rates of pet ownership and the evolving tourism behaviors in countries like Chile. By exploring the intentions and preferences of Chilean pet owners, this study seeks to contribute to a more comprehensive and culturally diverse understanding of pet-inclusive travel. In this context, the cultural value of pet tourism lies in the changing role of pets as integral members of the family unit, shaping social interactions and consumer behavior in leisure and travel decisions. This cultural shift is particularly relevant in Latin American societies, where the emotional and symbolic bonds with pets increasingly mirror those observed in other developed markets [6,14].

Therefore, this research seeks to analyze how these factors influence attitude, travel intention, and willingness to pay for pet-inclusive tourism in the Chilean context.

## 2. Literature Review

### 2.1. Intention to Travel with Pets

Understanding travel intentions and their impact on tourist behavior is a key focus of tourism research [19]. Travel intention is defined as a desire to travel or visit a specific tourist destination, evaluating the trip’s costs based on information collected by external opinions, either due to the location or the entertainment available [20,21]. Therefore, the stronger a person’s travel intention is, the greater the probability of traveling [19,22,23]. Previous research has found that pet owners take their pets when traveling for various reasons. These include the pet being loyal to them, the pet being considered a part of the family, the owner feeling safer in the pet’s presence, and the owner enjoying the pet’s companionship [11,16,18,24].

Traveling with pets, a relatively new and rapidly growing trend in the past decade, has marked a significant shift in travel patterns [11,13,25]. This shift can be attributed to the evolving perception of pets as integral family members, increasing the demand for pet-friendly travel options [25]. The concurrent rise in tourist establishments and services that cater to pets has further propelled this trend, making pet-inclusive travel more accessible and popular than ever before [7]. Previous research has indicated that emotional attachment, benefit, prestige, and novelty directly affect the intention to travel with pets [11,12,13]. These factors contribute to our understanding of data that helps create various value propositions to increase the participation of pets in tourist areas. However, the lack of attention to this market segment and the limited existing research on consumer perspectives regarding pets underscore the need for further investigation [11,12].

### 2.2. Conceptual Model and Research Hypothesis 

Based on the previous literature review, an integrated conceptual model is proposed to examine the relationships between the determining factors of pet tourism. The model considers four independent variables (emotional attachment, perceived benefits, prestige, and novelty) and their influence on attitude and intention to travel with pets. Additionally, it analyzes how these relationships affect the willingness to pay more for pet-friendly tourism services. Figure 1 presents the theoretical model and the proposed hypotheses that guide this research.

The model suggests that emotional attachment, perceived benefits, prestige, and novelty influence the intention to travel with pets directly and indirectly, with attitude mediating variables in these relationships. Additionally, it is proposed that attitude and travel intention are direct antecedents of the willingness to pay more for tourism services, including pets.

#### 2.2.1. Attitudes Toward Traveling with Pets

Attitude is a psychological state influenced by experience, which affects individual responses [26]. Attitudes refer to relatively stable beliefs that evaluate, describe, and recommend actions the individual considers in decision-making, which, while persistent over relevant periods, can evolve through experience and new information [27]. The relationship between attitudes toward and intention to travel with pets is critical in understanding consumer behavior in pet-friendly tourism [11]. There is a lack of comprehensive research on pet owners’ opinions about traveling with their furry companions and the factors influencing their decision to bring them on vacation [28].

Previous research has suggested that individuals with positive attitudes toward traveling with their pets are likelier to express an intention to do so [1,2,3]. This finding is consistent with the theory of planned behavior, which posits that attitudes are a significant predictor of behavioral intentions [29,30]. Researchers such as Mlakar & Korze [3] have found a strong link between attitude and behavior regarding pet ownership and travel. If someone has a positive attitude towards traveling with their pets, they are likelier to take trips that include their furry friends. People with a positive attitude toward traveling with pets tend to value pet-friendly amenities and services more. Examples of such amenities, particularly in the hotel sector, include designated pet-friendly rooms, welcome kits for pets, food and water bowls, in-room pet menus, and access to green areas or nearby trails [1]. This leads to a greater willingness to pay more for pet-inclusive travel options. Similarly, Zhang et al. [31] state that those with favorable attitudes toward pet-friendly accommodations are more likely to prioritize pet-related features and spend more money to ensure a comfortable and enjoyable experience for themselves and their pets. Based on the current literature, the following are hypothesized:

**Hypothesis 1 (H1):** 
*Attitudes toward traveling with pets directly and positively influence the intention to travel with pets.*


**Hypothesis 1a (H1a):** 
*Attitudes toward traveling with pets directly and positively influence the intention to pay more to travel with pets.*


#### 2.2.2. Emotional Attachment

Emotional attachment refers to a deep emotional bond that a person develops toward another person, an object, or even an idea [24,32]. It involves feelings of connection, affection, and often a sense of emotional dependence. This type of attachment can be formed through shared experiences, meaningful interactions, familiarity, and proximity to something or someone [2,33].

According to Kirillova et al. [12], emotional attachment in pet tourism is operationalized through separation-related indicators, measuring the distress, guilt, and worry pet owners experience when traveling without their companion animals. This approach reflects the premise that stronger emotional bonds manifest as greater separation anxiety, which may motivate owners to seek travel options that include their pets.

Previous research has found that pet owners take their pets when traveling for various reasons. These include the pet being loyal to them, being considered a part of the family, feeling safer in the pet’s presence, and enjoying the pet’s companionship [11,16,18]. The authors suggest that the more attached pet owners are to their pets, the more likely they are to consider pets family members [25]. Consequently, owners tend to include their pets in tourist activities [11,34]. In this context, pet owners are more likely to travel with their pets if they have a strong attachment to them [11,16]. Emotional attachment towards pets also significantly determines an individual’s intention towards traveling with their pets. According to March and Woodside [35], parents may modify their attitudes toward planning tourist activities with their children based on their level of attachment. Similarly, Peng et al. [16] found that attachment is a fundamental aspect of the relationship between pets and their owners. It directly influences owners’ attitudes toward engaging in activities with their pets [13]. Based on these findings, we propose the following hypotheses:

**Hypothesis 2 (H2):** 
*Emotional attachment to pets directly and positively affects attitudes toward traveling with pets.*


**Hypothesis 3 (H3):** 
*Emotional attachment to pets directly and positively affects the intention to travel with pets.*


#### 2.2.3. Perceived Benefits

Perceived benefits refer to an individual’s or society’s belief that they will benefit from a particular situation or circumstance [9,36,37]. It is a cognitive emotion that can positively influence people’s behavior [38]. In addition to being a cognitive–emotional construct, perceived benefits play a crucial role in shaping decision-making and the attitudes of individuals [9,11]. Owners’ perceived benefits to pets refer to owners’ beliefs about their pets’ well-being improvements, including happiness and physical benefits from exercise [12,13]. Studies have shown that many pet owners view their pets as part of their family and are willing to spend money on their pets’ well-being. This can include paying for costly veterinary treatments or purchasing high-end products for their pets [9,11,37].

Evidence has shown a positive correlation between pet ownership and the various benefits pets provide to their owners [11,12]. Many pet owners travel with their pets for enjoyment and to allow them to experience new surroundings, learn new things, and engage in exercise [10,13,18]. Additionally, owning pets can increase social interaction and reduce feelings of loneliness [39]. In this context, a relationship has been observed between pet ownership and the benefits promotion of pet owners when carrying out daily activities and even including them in recreational trips, confirming that pet owners perceive a benefit in the intention to travel with them [11,12].

The perceived benefits construct is defined following Kirillova et al. [12], who specify that perceived benefits in pet tourism refer to the advantages that pet owners perceive for themselves when traveling with their companion animals. These owner-focused benefits encompass enhanced vacation experiences, increased personal happiness, satisfaction with pet-friendly services, and emotional fulfillment, clarifying that the construct measures how traveling with pets improves the owner’s experience rather than focusing on benefits to the pets themselves. Rather than incorporating costs or net benefits, this focus on perceived benefits follows the established theoretical approach in the tourism literature that emphasizes positive motivational drivers and aligns with validated measurement scales in pet tourism research.

Based on what has been stated above, the following hypotheses are presented:

**Hypothesis 4 (H4):** 
*Perceived benefits to pets directly and positively affect their attitudes toward traveling with pets.*


**Hypothesis 5 (H5):** 
*Perceived benefits to pets directly and positively affect the intention to travel with pets.*


#### 2.2.4. Prestige

According to Achabou et al. [40], prestige refers to the desire to achieve a high status in the eyes of others. However, this interpretation can vary from person to person based on their socioeconomic background, social interactions, aspirations, quality of life, and political identity [41]. In this context, prestige can be understood as the perception of personal ability or achievement in a particular domain, often expressed through consumption choices [42]. It can also be seen as a symbol of exclusivity in different contexts, such as owning specific brands or possessing high-value items. Studies have shown that owning pets can provide a sense of status among pet owners, where people perceive their pets as a means of social recognition and associate their pets’ prestige with their quality of life [43]. People often seek to increase their social status by indulging their pets in luxury services or owning specific breeds [17]. As a noted by Veblen [44], pets can reflect social status through conspicuous consumption. Thus, the effect of prestige on travel attitudes may vary by species or breed, a point worth exploring in future research.

Previous studies have examined the correlation between pet ownership and social status [11,42]. For instance, Beverland et al. [17] suggest that some pet owners seek specific breeds or exotic pets to display their social status. Similarly, Xia et al. [41] found that pet owners often make consumption decisions based on status considerations and social identity. Tang et al. [11] also proposed that prestige significantly influences pet owners’ intention to travel with their pets. Therefore, we propose the following hypotheses:

**Hypothesis 6 (H6):** 
*Prestige associated with pet ownership directly and positively affects attitudes toward traveling with pets.*


**Hypothesis 7 (H7):** 
*Prestige associated with pet ownership directly and positively affects the intention to travel with pets.*


#### 2.2.5. Novelty

Novelty is defined as a desire to seek new and different experiences through leisure travel, motivated by the need to experience excitement, adventure, and surprise and relieve boredom [45,46,47]. In the context of pet tourism, novelty refers specifically to experiencing something significant for the first time when traveling with companion animals, encompassing the unique aspects of including pets in tourism experiences [11].

Previous research has explored how traveling with pets is perceived as innovative and novel, and many pet owners view traveling with their pets as a more exhilarating experience [11,13,18,48]. The novelty construct specifically measures the enjoyment, sense of novelty, and unique adventure derived from pet-inclusive tourism experiences rather than general destination novelty [12]. Additionally, Tang et al. [11] suggested that novelty encompasses a desire to encounter new experiences during trips, often driven by emotions such as thrill, adventure, surprise, and the alleviation of boredom; also, Hung et al. [48] suggested that novelty is associated with the joy of traveling with pets. In the same vein, it has been established within this research context that, for many pet owners, experiencing something unique and adventurous, such as traveling with pets, is highly enjoyable and novel [11,12,13]. Based on these findings, the following hypotheses are proposed:

**Hypothesis 8 (H8):** 
*The novelty of traveling with pets directly and positively affects attitudes toward traveling with pets.*


**Hypothesis 9 (H9):** 
*The novelty of traveling with pets directly and positively affects the intention to travel with pets.*


#### 2.2.6. Intention to Pay More to Travel with Pets

Intention to pay refers to the amount of money a person is willing to pay for a service, considering the product’s value [49,50,51]. The motivation behind tourists’ travel impacts their willingness to pay for a tourist service [52]. Previous research has shown a link between a tourist’s attitude towards traveling and their willingness to pay more for a service. A person’s attitude is cognitively and emotionally formed, which plays a significant role in their decision to travel and spend more money on it [53]. Similarly, the intention to pay for a tourism service is closely related to tourists’ motivation to travel and their behavioral intention [54]. In this context, Hultman et al. [53] affirmed that the intention to pay more to travel with pets is measured through one’s (1) willingness to have more expensive vacations when traveling with pets, (2) willingness to pay extra for better pet experiences that are guaranteed, and (3) willingness to pay more for pet-friendly accommodations compared to non-pet-friendly alternatives.

Moreover, recent studies have indicated that the increasing intention of deeply bonded pet owners to journey with their pets implies a willingness to pay additional funds for such experiences and extend the duration of their travel [1,7,12]. Many pet owners now feel it is essential to include their pets in their travel plans and activities. This has led to increased demand for tourism services catering to the needs and preferences of pets and their owners. As a result, pet owners intend to pay more to ensure their pets can participate in their travel experiences [7,55]. Based on the above, the following hypothesis is proposed:

**Hypothesis 10 (H10):** 
*The intention to travel with pets directly and positively influences the intention to pay more to travel with pets.*


## 3. Methodology

### 3.1. Sample

Data were collected through an online survey using Google Forms from September to November 2022 in Chile (See Appendix A). This study targeted people over 18 who are pet owners—specifically of cats and dogs—and intend to travel, go on vacation, or visit a specific destination in the next three months. Convenience sampling was used to reach the target audience, promoting participation in this study through forums and specialized Instagram accounts. In addition, participation in a drawing for a Gift Card with money for a pet store was offered; those who wished to participate did so by providing their email address. To eliminate ambiguities in the questionnaire, a pilot test was applied to 30 pet owners before the start of the field study. Although this preliminary test could not fully eliminate ambiguities, it helped reduce them. A descriptive analysis of the pilot data was conducted to identify items that led to comprehension issues or inconsistent responses. As a result, three items were slightly reworded to enhance clarity and ensure semantic consistency. The field study was conducted once the instrument’s parameters were verified and minor modifications were considered. To determine the sample size, G* Power software (v3.1.9.4.) was used along with an a priori test [56]. Using a power probability of 0.95, an effect size of 0.15 (mean), an alpha error probability of 0.05, and 7 predictors, the minimum sample size required was 153. Once the data collection was completed, 261 responses were collected; after invalid questionnaires were excluded, the final sample comprised 249 subjects. The post hoc analysis—maintaining effect size values of 0.15, a confidence interval of 0.05, and 7 predictors—achieved a power of 0.998, exceeding the minimum of 0.8 [56]. Of the participants, 66.7% were women and 33.3% were men. Most of the respondents were between 25 and 34 years old, and this range represented 37.8% of the sample. Regarding income, 32.9% said they earned more than three times the minimum monthly income. At the time of this study, the legal minimum monthly income in Chile was approximately CLP 460,000 (around USD 520), meaning this group earned roughly CLP 1,380,000 or more per month (about USD 1560). Finally, 76.7% of the participants had a university degree or professional qualification. The detailed sociodemographic characteristics of the sample can be found in Table 1.

### 3.2. Measurement Scale

The measurement scales for all constructs in our theoretical model were adapted from the established literature and evaluated using a 5-point Likert scale (1 = “completely disagree” to 5 = “completely agree”). Emotional attachment was measured using three items, while perceived benefits were assessed through four items, both adapted from Kirillova et al. [57]. Following Tang et al. [11], we measured prestige and novelty using three items each and intention to travel with pets using three items. Attitudes toward traveling with pets were evaluated using four indicators developed by Peng et al. [16]. Additionally, the intention to pay more for pet travel was measured using three items proposed by Hultman et al. [53].

### 3.3. Analysis Technique

Partial least squares (PLS) modeling was performed using SmartPLS 4.0 as a statistical tool to examine the measurement and structural models [58]. This approach is suitable for modeling latent variables without strict standard distribution requirements. Hair et al. [59] have suggested that specific indicators be calculated to assess the validity and reliability of the measurement and structural models.

## 4. Results

### 4.1. Measurement Model

First, we conducted Harmon’s one-way analysis of variance to assess any potential common method bias. The findings indicated a common variance of 45%, below the critical threshold of 50% established by Podsakoff et al. [60]. Therefore, no significant concerns about common method bias were identified in our data. We then assessed the measurement model before estimating the structural equation of the model to test our hypothesis. The results of the psychometric properties of the model can be viewed in Table 2. The reliability indicators yielded good results, evident from the fact that all the values of Cronbach’s Alpha and Composite Reliability were above the recommended value of ≥0.70 [61]. Regarding the additional Dillon–Goldstein Rho Reliability indicator, we confirmed that all Rho values are above 0.7 [62,63].

The AVE values were examined to assess convergent validity, confirming that they all meet the minimum accepted level of 0.5 [64]. Finally, as stated by Hair et al. [59], the reliability of each variable was assessed using the loading indicator. To be considered part of a variable, an indicator must have a loading ≥ 0.70 or at least 0.50 when the instrument is used in other contexts [64]. All indicators meet this criterion. Therefore, the results show good internal consistency of the construct and adequate convergent validity.

Fornell and Larcker’s criterion assessed discriminant validity. This criterion indicates that the AVE square root must be higher than the correlations with the other variables [65]. As Table 3 shows, the correlations were lower than all the square roots of the AVE and are located below the diagonal, while the AVE square roots themselves are located on the diagonal. Hair et al. recently proposed the HTMT ratio method [59], and Henseler et al. [61] were also applied to assess discriminant validity. HTMT establishes that the maximum appropriate value is 0.90 [63]. Additionally, Table 3 shows that all the HTMT ratio values above the diagonal are below 0.90. Therefore, the results obtained reveal that discriminant validity does indeed exist.

### 4.2. Structural Model

The structural model was estimated after evaluating the measurement model’s psychometric properties. First, the Standardized Root Mean Residual (SRMR) was assessed. It measures the standardized difference between the observed and predicted correlation and is an absolute fit measure. Henseler et al. [61] describe SRMR as a goodness-of-fit measure for PLS-SEM that can be used to avoid misspecifications of the model. A value below 0.08 indicates a good fit [65]. The SRMR yielded a value of 0.067, which provides a good fit for the proposed model. Additionally, it was used for the bootstrapping procedure on 5000 subsamples to evaluate the causal relationships and their level of significance [64]. Finally, to confirm the structural model’s predictive capacity, the R2 values were evaluated. According to Falk and Miller [57], R2 values should be higher than 0.1. Lower values, even if significant, would not be acceptable.

Table 4 shows the goodness of fit of the model (SRMR), the result of the hypotheses proposed in the structural model (path coefficients and *p*-value), and the predictive capacity of the model (R2). As a result, the hypothesis testing shows that H1, H4, H5, H7, H8, H10, and H1a are accepted. Meanwhile, H2, H3, H6, and H9 are rejected.

## 5. Discussion

### 5.1. Influence of EAT, PB, PR, and NOV in Attitudes Toward Traveling with Pets (ATP)

The support for Hypotheses 4 and 8 shows that both perceived benefits and novelty positively impact pet owners’ attitudes toward traveling with their pets. These results align with previous research; studies by Tang et al. [11] and Chong, Choong, and Sam [66] indicate that owners who see benefits for their pets when traveling or seeking new and stimulating experiences are more willing to include their pets in their travel plans. Additionally, the acceptance of H8 confirms that novelty concerning pets positively influences attitudes toward traveling with them [11,18,48]. These findings emphasize the importance of these factors as critical motivators in shaping an attitude conducive to pet-friendly tourism.

However, the rejection of Hypotheses 2 and 6 suggests that neither emotional attachment nor prestige associated with pet ownership significantly affects attitudes toward traveling with pets. In the case of H2, the studies by Kirillova et al. [12] and Tang et al. [11] support the idea that emotional bonding with pets can foster a positive attitude towards traveling with them. However, these studies were carried out in the US and China, and in the case of Chile, this link does not seem to have a direct and positive impact on said attitudes. Factors such as cultural differences and ease of travel can affect people’s relationships with their pets. In Chile, traveling with pets may not be culturally valued as much, regardless of the degree of emotional attachment. Regarding H6, the perception of prestige associated with pets does not strongly influence attitudes toward traveling with them. This aligns with the findings of Tang et al. [11], who noted that prestige does not directly affect the intention to travel with pets. Instead, owners consider socialization, novelty, and perceived benefits.

### 5.2. Influence of EAT, PB, PR, NOV, and ATP in the Intention to Travel with Pets

The acceptance of hypotheses H1, H5, and H7 highlights the importance of the pet owner’s positive attitude toward traveling with their pet, perceived benefits, and prestige in the intention to travel with them. These findings are consistent with prior studies, such as that of Ying et al. [13], which demonstrated that, although perceived restrictions may inhibit traveling with pets, a positive attitude can overcome these barriers and encourage travel intention. Furthermore, Chong et al. [66] suggested that the relationship between owners and their pets, including the recognition of benefits and prestige, can influence the intention to travel with them. Therefore, the results highlight the significance of these characteristics as crucial drivers in influencing an intention that promotes pet-friendly travel.

On the other hand, the cultural and geographical differences between China and Chile could be why Hypotheses 3 (H3) and 9 (H9) were rejected. The study by Tang et al. [11] was conducted in China, where attitudes and behaviors toward pets and pet travel policies and regulations may differ from those in Chile. These differences could have influenced the relationship between emotional attachment, novelty toward pets, and the intention to travel with them. Therefore, although these hypotheses were supported in the Chinese study, sufficient evidence was not found to support them in the Chilean context.

### 5.3. Influence of ATP and ITP in the Intention to Pay More to Travel with Pets (IPM)

The empirical support for Hypotheses 10 and 1a strengthens the notion that the intention and a favorable attitude towards traveling with pets are crucial determinants that impact the willingness to spend more money on such journeys. The academic literature supports this relationship; for example, studies have shown that behavioral intention, such as the intention to travel with pets, can predict the willingness to incur additional expenses to improve one’s travel experience [1,7,12]. Furthermore, a positive attitude towards traveling with pets has been associated with a greater willingness to pay more to travel with them [1]. These findings suggest that pet owners value the inclusion of their animal companions in their travel experiences and are willing to invest in their well-being and comfort.

## 6. Conclusions

The hypothesis test reveals an interesting pattern in the intention to travel with pets in Chile. Seven hypotheses (H1, H4, H5, H7, H8, H10, and H1a) are confirmed, showing that attitudes towards traveling with pets, perceived benefits, prestige, and novelty determine travel intention and willingness to pay more. Specifically, positive attitudes (H1) directly influence travel intention, while perceived benefits (H4, H5) and novelty (H8) significantly impact attitudes and intention. Prestige (H7) was shown to be relevant for travel intention, and both attitude and intention (H10, H1a) are crucial for willingness to pay more.

On the other hand, four hypotheses (H2, H3, H6, and H9) are rejected, suggesting that emotional attachment does not significantly influence attitude or travel intention (H2, H3), prestige does not affect attitude (H6), and novelty does not directly impact travel intention (H9). These results contrast with previous studies conducted in other countries, which could be attributed to cultural and contextual differences specific to the Chilean market.

### 6.1. Limitations of This Study

This study faces certain limitations that must be considered when interpreting its results. This research is limited exclusively to the Chilean context, which restricts the generalization of the findings to other countries or cultures. In addition, as it is a cross-sectional study, the data reflect a specific moment in time without capturing possible temporal variations in the attitudes and behaviors of pet owners. Based on online surveys, the methodology could have introduced a bias towards users with greater access and familiarity with digital media. Likewise, although significant variables were analyzed, other potentially relevant factors were not included in the model, such as the available pet-friendly tourism infrastructure or the current legal restrictions for the transportation of pets. In addition, the sample was composed predominantly of individuals under 34 years of age (72%) and single respondents (96%). These demographic characteristics may influence the motivation and flexibility to travel with pets.

### 6.2. Implications

This research contributes to the existing literature on pet-friendly tourism by validating theoretical relationships in the Latin American context, specifically in Chile. The results expand the understanding of the theory of planned behavior by identifying that perceived benefits and novelty are more significant determinants than emotional attachment in this cultural context. This study also contributes to knowledge about willingness to pay for tourist services, establishing a clear relationship between positive attitudes and the intention to invest more in pet-friendly experiences.

The findings provide valuable guidelines for the tourism sector in terms of business implications. Companies should prioritize the development of value propositions focused on tangible benefits and novel experiences rather than on emotional connections. Implementing differentiated pricing strategies supported by services that justify the added value is recommended. The hotel and transportation sector can use these results to design pet-friendly services that emphasize comfort and unique experiences. Tourism marketing agencies should focus on the practical benefits and innovative elements of traveling with pets, adapting their messages to the Chilean cultural context.

### 6.3. Future Research

Expanding this research internationally, particularly in Latin America, is recommended to establish cultural comparisons on tourist behavior with pets. Longitudinal studies would be valuable in understanding the evolution of attitudes and behaviors over time. Future research should also examine how demographic factors such as age and marital status may moderate key relationships in pet tourism behavior. Given that our sample consisted mainly of young and single individuals, these characteristics could meaningfully shape travel motivations and constraints. Multi-group analysis or interaction effects could be applied to assess these moderating roles. It is important to explore additional variables, such as the impact of pet-friendly tourism infrastructure, transportation regulations, and the role of emerging technologies in service delivery. Future research should examine specific segments by pet type and destination and the influence of sociodemographic factors on decision-making. Finally, it would also be relevant to study the relationship between pet-friendly tourism and sustainability by considering environmental and social aspects.

## Figures and Tables

**Figure 1 animals-15-01741-f001:**
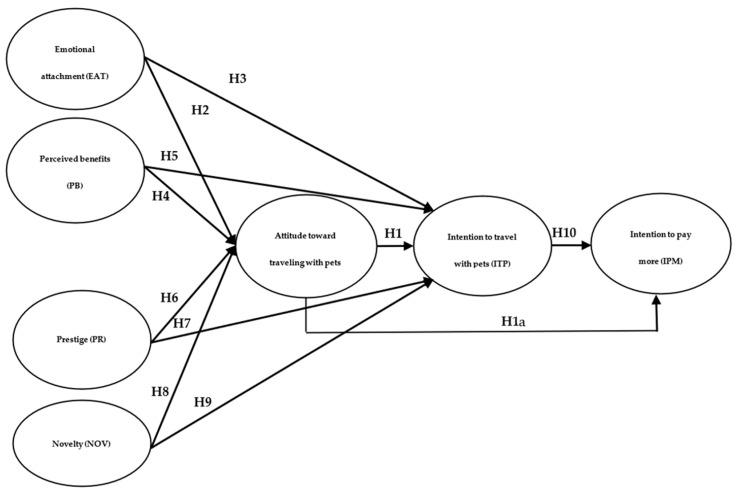
Proposed model of travel intentions with pets.

**Table 1 animals-15-01741-t001:** Sociodemographic data of the sample (*n* = 249).

Category		Frequency	Percentage
Age range	18–24	87	35.0%
	25–34	94	37.8%
	35–44	25	10.0%
	45–54	22	8.8%
	Over 54	21	8.4%
Sex	Male	83	33.3%
	Female	166	66.7%
Civil status	Single	394	96.1%
	Married	16	3.9%
Education	School	28	11.2%
	University (undergraduate)	191	76.7%
	University (postgraduate)	30	12%
Family income	Up to 1 minimum wages	61	24.5%
	From 1 to 2 minimum wages	55	22.0%
	From 2 to 3 minimum wages	51	20.4%
	Greater than 3 minimum wages	82	32.9%

**Table 2 animals-15-01741-t002:** Reliability and validity of the measurement model.

Variable	Item	Factor Loading	Cronbach Alpha	Rho_a	Composed Reliability (CR)	Average Variance Extracted (AVE)
Emotional attachment (EAT)	EAT1	0.911	0.890		0.931	0.819
	EAT2	0.914		0.904		
	EAT3	0.889				
Perceived benefits (PB)	PB1	0.825				
	PB2	0.655				
	PB3	0.805	0.885	0.904	0.912	0.636
	PB4	0.773				
	PB5	0.868				
	PB6	0.873				
Prestige (PR)	PR1	0.631				
	PR2	0.881	0.844	0.896	0.893	0.680
	PR3	0.909				
	PR4	0.872				
Novelty (NOV)	NOV1	0.913				
	NOV2	0.895	0.924	0.926	0.946	0.814
	NOV3	0.919				
	NOV4	0.880				
Attitudes toward traveling with pets (ATP)	ATP1	0.943				
	ATP2	0.937				
	ATP3	0.940	0.964	0.964	0.972	0.875
	ATP4	0.936				
	ATP5	0.919				
Intention to travel with pets (ITP)	ITP1	0.883				
	ITP2	0.932	0.879	0.879	0.925	0.805
	ITP3	0.877				
Intention to pay more (IPM)	IPM1	0.927				
	IPM2	0.954	0.931	0.932	0.956	0.879
	IPM3	0.932				
	IPM4	0.872				

**Table 3 animals-15-01741-t003:** Discriminant validity assessed through Fornell–Larcker criterion and HTMT ratio, including AVE square roots and inter-construct correlations.

	EAT	PB	PR	NOV	ATP	ITP	IPM
Emotional attachment (EAT)	0.905	0.692	0.573	0.634	0.568	0.354	0.602
Perceived benefits (PB)	0.622	0.798	0.704	0.839	0.833	0.541	0.812
Prestige (PR)	0.521	0.637	0.825	0.785	0.681	0.506	0.702
Novelty (NOV)	0.581	0.773	0.719	0.902	0.835	0.493	0.816
Attitudes toward traveling with pets (ATP)	0.532	0.784	0.644	0.791	0.935	0.541	0.807
Intention to travel with pets (ITP)	0.318	0.492	0.462	0.445	0.499	0.897	0.520
Intention to pay more (IPM)	0.554	0.744	0.644	0.758	0.765	0.470	0.937

Note: Fornell and Larcker diagonal; HTMT above the diagonal; correlational values below the diagonal.

**Table 4 animals-15-01741-t004:** Assessment of direct effects and hypothesis testing.

Hypothesis	Relationship	Path Coefficients	*p*-Values	Results
H1	ATP→ITP	0.247	0.016	Accepted *
H2	EAT→ATP	−0.013	0.798	Rejected
H3	EAT→ITP	−0.032	0.700	Rejected
H4	PB→ATP	0.416	0.000	Accepted ***
H5	PB→ITP	0.231	0.045	Accepted *
H6	PR→ATP	0.089	0.118	Rejected
H7	PR→ITP	0.223	0.013	Accepted *
H8	NOV→ATP	0.413	0.000	Accepted ***
H9	NOV→ITP	−0.071	0.571	Rejected
H10	ATP→IPM	0.706	0.000	Accepted ***
H1a	ITP→IPM	0.118	0.015	Accepted *

R2(ATP) = 0.698; R2(ITP) = 0.284; R2(IPM) = 0.593; SRMR = 0.067. * *p* < 0.05; *** *p* < 0.001.

## Data Availability

Data can be requested by writing to the corresponding author of this publication.

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
