# Peer review of "Determinants of Pet-Friendly Tourism Behavior: An Empirical Analysis from Chile"

_animals, 2025, doi:10.3390/ani15121741_

Round 1
Reviewer 1 Report
Comments and Suggestions for Authors
Thank you for the opportunity to review this interesting study. In general, I believe it has sufficient merit to be published eventually, but there were some gaps that I would like to see addressed. Both the findings and their tabular presentation were very "clean," which provides an advantage for a study such as this. Below I offer suggestions to strengthen the manuscript.
- The phrase "pet tourism" in the title and appearing in the text could be confusing to some readers, especially those accustomed to reading about tourism studies. Tourism industries have become incredibly segmented, of course, with a myriad of specialty marketing niches, e.g., dark tourism, solar eclipse tourism, extreme tourism, etc. The specialty's name is typically based on the desired activity/event/experience, so "pet tourism" might suggest travel to browse for, adopt, or buy a pet. I prefer other terms that the authors use elsewhere in the text such as "pet-inclusive travel," "pet accompanying," or "pet friendly" tourism.
- Chile should probably appear in the title itself given the cultural contexts discussed by the authors, e.g., for the unexpected findings regarding emotional attachment.
- The reader would benefit from being able to view a list of indicators for each latent variable in the model; these could be included in a separate table that also provides descriptive statistics for each indicator. By the way, why not give the SEM a name in Figure 1? Further for Figure 1, I would suggest that the authors include the 3-letter label that is used for each latent variable when reporting findings in Tables 2-4.
- In terms of inclusion/exclusion criteria, were pets defined as being of any particular species? Were fish ruled out? If there were not exclusion criteria, did the study collect data on which species of pets were considered by survey respondents when responding? Was information collected on the number of pets? And how were respondents to answer if they had multiple pets -- in other words, were the questions particular to a specific pet?
- Lines 60-61 provide an example of a pet-friendly tourism policy/service; could the authors provide a few more examples to illustrate what could make travel more pet-friendly? Specifically, are most pet-related tourism services (particularly in Chile) about boarding, grooming, exercise, or socializing? Lines 135-137 also mention "pet-friendly amenities and services" -- what are some examples?
- In Lines 123-124, the authors note that "Attitudes refer to enduring beliefs that..." Attitudes can change, so how "enduring" is this meant to be?
- Figure 1: It was unclear (particularly in Section 2.2.3) whether "perceived benefits" were particular to the pet, the human(s), or both. Should the model also incorporate (nonfinancial) "costs"? Or perhaps something like "net" benefits?
- The hypotheses involving emotional attachment involve positive effects on dependent variables. Is it possible that certain types of attachment or their strengths could have negative effects on those same endogenous variables?
- Figure 1: To clarify, is novelty not so much about the destination or experiences in general. Is novelty only about aspects associated with including the pet(s) in the tourism experience? This would be made clearer for the reader if the specific indicators were included somewhere, of course
- Figure 1: How was "Intention to pay more" measured? Were the statements to the effect of being willing to pay more include their pet(s)?
- Line 326: It would help to clarify for the reader that correlations appeared below the diagonal.
- Lines 378-379: "Pet's positive attitude" would seem to have a different meaning than the authors likely intended.
- Could the effect of prestige on attitude to travel be conditional on species and/or breed? Thorstein Veblen's classic book, The Theory of the Leisure Class, could be cited on conspicuous consumption, conspicuous leisure, and vicarious consumption via pet's leisure. As the authors may know, Veblen specifically mentioned pets as examples to animate the concept of invidious comparison.
- It is unfortunate that the data and model didn't capture whether the intended travel occurred.
- Lines 264-265: Pretesting with 30 pet owners will not likely "eliminate ambiguities" with within a survey instrument. It may, however, help to reduce ambiguities. Which analyses were done with the pilot data? Which specific "minor modifications" were implemented?
- Lines 276: What do "three minimum monthly incomes" signify?
- Lines 300-302: There may be a missing word to two in this sentence.
- Line 342: Is "On the other hand" an appropriate way to begin this sentence?
Author Response
Dear Reviewer 1,
We extend our sincere gratitude for your insightful comments, which have been invaluable in enhancing the quality of our manuscript. Your thoughtful feedback has contributed significantly to refining our work, and we have made concerted efforts to address each of your suggestions. We are optimistic that this revised version of the paper now meets the anticipated standards for publication in this esteemed journal. Below is a comprehensive list of responses addressing your comments and suggestions. Thank you once again for your time and expertise.
OBSERVATION 1: Thank you for the opportunity to review this interesting study. In general, I believe it has sufficient merit to be published eventually, but there were some gaps that I would like to see addressed. Both the findings and their tabular presentation were very "clean," which provides an advantage for a study such as this. Below I offer suggestions to strengthen the manuscript.
RESPONSE 1: We sincerely appreciate your thorough review and constructive feedback. We are pleased that you found the study's merit and the clarity of our findings and presentation valuable. We have carefully addressed all the gaps and suggestions you identified in your review, as detailed in our point-by-point responses below. These revisions have significantly strengthened the manuscript and enhanced its contribution to the pet-friendly tourism literature.
OBSERVATION 2: The phrase "pet tourism" in the title and appearing in the text could be confusing to some readers, especially those accustomed to reading about tourism studies. Tourism industries have become incredibly segmented, of course, with a myriad of specialty marketing niches, e.g., dark tourism, solar eclipse tourism, extreme tourism, etc. The specialty's name is typically based on the desired activity/event/experience, so "pet tourism" might suggest travel to browse for, adopt, or buy a pet. I prefer other terms that the authors use elsewhere in the text such as "pet-inclusive travel," "pet accompanying," or "pet friendly" tourism. Chile should probably appear in the title itself given the cultural contexts discussed by the authors, e.g., for the unexpected findings regarding emotional attachment.
RESPONSE 2: We appreciate these valuable observations about both terminology clarity and geographical specificity. You are absolutely correct that "pet tourism" could be ambiguous and potentially misleading, and that Chile should be prominently featured given the cultural context and unexpected findings. We have revised the title to: "Determinants of Pet-Friendly Tourism Behavior in Chile: A Behavioral Analysis." This revision addresses both concerns by using clearer terminology that better reflects tourism services that welcome and accommodate pets, while incorporating the geographical and cultural context that is central to our findings. We will consistently use "pet-friendly tourism" throughout the manuscript to eliminate potential confusion and align with established tourism literature conventions.
OBSERVATION 3: The reader would benefit from being able to view a list of indicators for each latent variable in the model; these could be included in a separate table that also provides descriptive statistics for each indicator. By the way, why not give the SEM a name in Figure 1? Further for Figure 1, I would suggest that the authors include the 3-letter label that is used for each latent variable when reporting findings in Tables 2-4.
RESPONSE 3: In response, we have now included a detailed table listing all indicators associated with each latent variable, along with their corresponding descriptive statistics. This table has been added as Appendix A – Table 1A. Additionally, we have revised the title of Figure 1 to better reflect the content and purpose of the model. The new title is: "Proposed Model of Travel Intention with Pets." We have also incorporated the two or three-letter labels for each latent variable into Figure 1, consistent with those used in Tables 2–4. All changes have been highlighted in blue in the revised manuscript for ease of review.
OBSERVATION 4: In terms of inclusion/exclusion criteria, were pets defined as being of any particular species? Were fish ruled out? If there were not exclusion criteria, did the study collect data on which species of pets were considered by survey respondents when responding? Was information collected on the number of pets? And how were respondents to answer if they had multiple pets -- in other words, were the questions particular to a specific pet?
RESPONSE 4: In terms of inclusion/exclusion criteria, pets were implicitly defined through a screening question that specifically inquired whether the respondent had at least one dog or cat. This filter ensured that the study focused only on individuals with these types of pets, as reflected in the revised manuscript (see text highlighted in blue). Other species, such as fish, were not explicitly ruled out but were effectively excluded through the design of this initial question. The study did not collect detailed information on other pet species beyond dogs and cats, nor did it gather data on the total number of pets owned by respondents. Regarding respondents with multiple pets, the survey questions referred to their general experience of traveling with pets rather than focusing on one specific animal. Therefore, participants were instructed to respond based on their overall experience with their pets.
OBSERVATION 5: Lines 60-61 provide an example of a pet-friendly tourism policy/service; could the authors provide a few more examples to illustrate what could make travel more pet-friendly? Specifically, are most pet-related tourism services (particularly in Chile) about boarding, grooming, exercise, or socializing? Lines 135-137 also mention "pet-friendly amenities and services" -- what are some examples?
RESPONSE 5: In response to your first comment regarding lines 60–61, we have expanded the paragraph to include additional examples that illustrate what makes travel more pet-friendly. As noted in the revised version (highlighted in blue), examples of pet-friendly services and destinations now include restaurants, airlines, hotels, and resort centers. Additionally, based on Taillon et al. [1], we emphasize the diversity of pet-related services within the tourism and hospitality sector, which go beyond boarding and grooming. Regarding your second comment on lines 135–137, we have added the following clarification to illustrate what is meant Examples of such amenities, particularly in the hotel sector, include designated pet-friendly rooms, welcome kits for pets, food and water bowls, in-room pet menus, and access to green areas or nearby trails. This addition is now included in the revised manuscript and has been highlighted in blue for your convenience.
OBSERVATION 6: In Lines 123-124, the authors note that "Attitudes refer to enduring beliefs that..." Attitudes can change, so how "enduring" is this meant to be?
RESPONSE 6: In this context, the term "enduring" refers to the relative stability of attitudes compared to momentary emotions or fleeting thoughts, rather than implying permanent unchangeability. Following Ajzen's Theory of Planned Behavior and established attitude research, attitudes are considered relatively stable psychological constructs that persist over periods relevant for decision-making processes, while still being subject to modification through new experiences, information, or social influences. We have revised lines 123-124 to: "Attitudes refer to relatively stable beliefs that evaluate, describe, and recommend actions the individual considers in decision-making, which while persistent over relevant periods, can evolve through experience and new information [27]." This revision better reflects attitudes' dynamic yet stable nature in behavioral research.
OBSERVATION 7: Figure 1: It was unclear (particularly in Section 2.2.3) whether "perceived benefits" were particular to the pet, the human(s), or both. Should the model also incorporate (nonfinancial) "costs"? Or perhaps something like "net" benefits?
RESPONSE 7: We have added an explicit definition within this section that specifies that "perceived benefits" refers exclusively to the advantages that pet owners perceive for themselves when traveling with their companion animals, following the conceptualization of Kirillova, Lee, and Lehto (2015). The new text clarifies that this construct measures the owner's perception of how traveling with pets improves their travel experience, including aspects such as enhanced vacation experiences, increased personal happiness, and satisfaction with pet-friendly services, distinguishing it from benefits directed toward the pets. This modification directly addresses your concerns about the construct's conceptual clarity while maintaining our framework's theoretical coherence.
OBSERVATION 8: The hypotheses involving emotional attachment involve positive effects on dependent variables. Is it possible that certain types of attachment or their strengths could have negative effects on those same endogenous variables?
RESPONSE 8: While we acknowledge that certain attachment types could have negative effects, our study focuses on how including pets in travel can contribute to strengthening healthy emotional bonds. The emotional attachment construct, operationalized through separation-related indicators following Kirillova et al. (2015), measures the distress owners experience when separated from their pets. Our positive hypotheses (H2 and H3) are grounded in the premise that pet-inclusive tourism serves as a solution to separation anxiety, potentially enhancing the human-pet bond through shared travel experiences. Rather than examining problematic attachment patterns, this research explores how the tourism industry can facilitate positive attachment outcomes by enabling owners to maintain proximity to their companion animals during leisure activities. This perspective aligns with the growing recognition of pets as family members and the industry's response to accommodate this relationship.
OBSERVATION 9: Figure 1: To clarify, is novelty not so much about the destination or experiences in general. Is novelty only about aspects associated with including the pet(s) in the tourism experience? This would be made clearer for the reader if the specific indicators were included somewhere, of course.
RESPONSE 9: You are correct in your interpretation. The novelty construct in our study refers specifically to the novel aspects of including pets in tourism experiences, not general destination novelty. We have clarified this in Section 2.2.5 by specifying that novelty measures the unique experiences, enjoyment, and sense of adventure derived specifically from pet-inclusive tourism rather than general travel novelty. The scale items focus on gaining enjoyment/pleasure, experiencing novelty, and having unique adventures, specifically in the context of traveling with companion animals. This distinction is important as it differentiates between destination-based novelty and the novelty of the pet-inclusive tourism experience itself. The revised text establishes this conceptual boundary to avoid confusion with general tourism novelty constructs.
OBSERVATION 10: Figure 1: How was "Intention to pay more" measured? Were the statements to the effect of being willing to pay more include their pet(s)?
RESPONSE 10: The "Intention to pay more" construct specifically includes references to pets in all measurement items. The scale was adapted from Hultman et al. (2015) and modified for pet tourism context. The three items measure: (1) willingness to have more expensive vacations when traveling with pets, (2) willingness to pay extra if it guarantees a better experience for the pet, and (3) willingness to pay more for pet-friendly accommodations compared to non-pet-friendly ones. All statements explicitly reference the pet context, ensuring that the willingness to pay more is directly related to pet-inclusive tourism services rather than general tourism expenditure. This adaptation ensures construct validity by maintaining the focus on pet-related tourism expenses.
OBSERVATION 11: Line 326: It would help to clarify for the reader that correlations appeared below the diagonal.
RESPONSE 11: As requested, we have clarified in line 326 that the correlations appear below the diagonal and that the AVE square roots are presented on the diagonal. This addition has been highlighted in blue in the revised manuscript.
OBSERVATION 12: Lines 378-379: "Pet's positive attitude" would seem to have a different meaning than the authors likely intended. Could the effect of prestige on attitude to travel be conditional on species and/or breed? Thorstein Veblen's classic book, The Theory of the Leisure Class, could be cited on conspicuous consumption, conspicuous leisure, and vicarious consumption via pet's leisure. As the authors may know, Veblen specifically mentioned pets as examples to animate the concept of invidious comparison.
RESPONSE 12: Regarding the first point, we agree that the phrase “pet’s positive attitude” could be misinterpreted. To improve clarity, we have revised the sentence to explicitly state “the pet owner’s positive attitude toward traveling with their pet”, which more accurately conveys the intended meaning. As for the second suggestion, we thank the reviewer for encouraging us to consider Thorstein Veblen’s contributions. We have now incorporated the following sentence into the Prestige section of the manuscript to reflect this perspective: “As noted by Veblen (1899), pets can reflect social status through conspicuous consumption. Thus, the effect of prestige on travel attitudes may vary by species or breed, a point worth exploring in future research.” This addition allows us to acknowledge the symbolic and social-signaling function of pet ownership, while also highlighting an important direction for future studies.
OBSERVATION 13: It is unfortunate that the data and model didn't capture whether the intended travel occurred. Lines 264-265: Pretesting with 30 pet owners will not likely "eliminate ambiguities" with within a survey instrument. It may, however, help to reduce ambiguities. Which analyses were done with the pilot data? Which specific "minor modifications" were implemented? Lines 276: What do "three minimum monthly incomes" signify? Lines 300-302: There may be a missing word to two in this sentence. Line 342: Is "On the other hand" an appropriate way to begin this sentence?
RESPONSE 13: We acknowledge that the data and model do not confirm whether the intended travel actually occurred. As described in the methodology section, the data were collected from individuals over the age of 18 who owned a dog or cat and indicated an intention to travel, go on vacation, or visit a specific destination within the next three months. The objective of the study was to examine behavioral intentions, not actual travel behavior. As noted, the preliminary testing with 30 pet owners was not intended to fully eliminate all potential ambiguities, but rather to help reduce them. Based on the pilot data, we conducted a descriptive analysis to identify items with comprehension issues or inconsistent responses. Minor modifications were made to the wording of three items to improve clarity and ensure semantic consistency. These adjustments were incorporated into the final version of the instrument.
We have added the word "specific" to clarify the sentence, as it was indeed missing. This correction has been highlighted in blue in the revised manuscript. The expression “three minimum monthly incomes” refers to three times the legal monthly minimum wage in Chile. At the time of the study, the minimum monthly income was approximately $460,000 CLP (Chilean pesos). Therefore, this category includes participants with an income of approximately $1,380,000 CLP or more per month. As requested, we have now added a brief explanation in the manuscript, clarifying this amount both in Chilean pesos and in U.S. dollars. This addition has been highlighted in blue. We have revised the sentence in lines 300–302 to improve clarity. Specifically, we added the word "specific" to complete the sentence and ensure its proper meaning. The corrected sentence now reads: Hair et al. [59] suggested that specific indicators were calculated to assess the validity and reliability of the measurement and structural models. This correction has been highlighted in blue in the revised manuscript. As suggested, we have replaced the expression "On the other hand" with a more appropriate academic connector. This change has been highlighted in blue in the revised manuscript.
Reviewer 2 Report
Comments and Suggestions for Authors
Animals
This is an interesting study about a more complex relationship. SEM is approriate, an based on a conceptual model. I have only a few small comments.
LINE 51. Here the term pet tourism is laid out. Is it possible to give a definition of pet tourism at this place?
Line 60. Full-stop is missing
Line 71ff: this might not be necessary. Probably this was requested from e reviewer of a previous journal (a leisure or tourism journal), but for Animals it might not ne necessary.
I think the theoretical background is developed very well and needs no further improvement, hypotheses are clearly derived from a conceptual model.
Line 321: perhaps write “factor loading” rather than loading factor?
The table captions could go a bit more into detail. Especially table 3
Line 336. Please consider to move some of the facts presented here into the methods. Use only the text/values where you compare your data, and not which cut-offs have been chosen, These could be better placed in the methods section.
I dn*t know what “source: own elaboration” means? Perhaps it is not necessary?
Perhaps consider other factors to present in addition to SRMR, for example RMSEA and CFI,
Line 459: I would delete the part of the pandemic.
Author Response
Dear Reviewer 2,
Thank you very much for your comments on our article. On behalf of our research group, we have corrected everything you requested. Below are the comments and their respective responses:
OBSERVATION 1: This is an interesting study about a more complex relationship. SEM is approriate, an based on a conceptual model. I have only a few small comments.
RESPONSE 1: Thank you very much for your positive feedback. We appreciate your recognition of the relevance of the topic and the appropriateness of using SEM to examine the proposed conceptual model. We have carefully addressed your comments and suggestions, which we believe have strengthened the clarity and overall quality of the manuscript.
OBSERVATION 2: LINE 51. Here the term pet tourism is laid out. Is it possible to give a definition of pet tourism at this place?
RESPONSE 2: Thank you for your helpful comment. We agree that defining the term pet tourism at the point where it is first introduced improves clarity for the reader. Accordingly, we have added the following definition to the manuscript (highlighted in blue): “Pet tourism refers to travel activities in which pet owners are accompanied by their pets, integrating the pets’ presence into the overall tourism experience. This concept involves not only the logistical aspects of traveling with animals but also the emotional, social, and psychological motivations that drive owners to include their pets in leisure travel.” Tang et al. (2022).
OBSERVATION 3: Line 60. Full-stop is missing
RESPONSE 3: Thank you for pointing this out. The missing full stop at line 60 has been added. The correction is highlighted in blue in the revised manuscript.
OBSERVATION 4: Line 71ff: this might not be necessary. Probably this was requested from e reviewer of a previous journal (a leisure or tourism journal), but for Animals it might not ne necessary.
RESPONSE 4: Thank you for your observation. We understand that the paragraph beginning at line 71 may not be essential for the scope of Animals. Therefore, we have proceeded to remove it from the revised version of the manuscript as suggested.
OBSERVATION 5: I think the theoretical background is developed very well and needs no further improvement, hypotheses are clearly derived from a conceptual model.
RESPONSE 5: We appreciate this positive feedback regarding the theoretical framework and hypothesis development. While we agree that the overall conceptual foundation is solid, we have made minor clarifications in response to specific observations from other reviewers to enhance clarity and precision in the construct definitions and operationalizations.
OBSERVATION 6: Line 321: perhaps write “factor loading” rather than loading factor?
RESPONSE 6: Thank you for your suggestion. We have replaced the term “loading factor” with “factor loading” to ensure correct terminology. This change has been highlighted in blue in the revised manuscript.
OBSERVATION 7: The table captions could go a bit more into detail. Especially table 3
RESPONSE 7: Thank you for your helpful comment. In response to your suggestion, we have revised the table captions to provide more descriptive and informative content, particularly for Table 3. The updated caption now clarifies that it includes both the Fornell–Larcker criterion and HTMT ratio, along with AVE square roots and inter-construct correlations. This change has been highlighted in blue in the revised manuscript.
OBSERVATION 8: Line 336. Please consider to move some of the facts presented here into the methods. Use only the text/values where you compare your data, and not which cut-offs have been chosen, These could be better placed in the methods section.
RESPONSE 8: After careful consideration, we have decided to retain the information in its current location, as it reflects the standard practice for reporting results in Partial Least Squares (PLS) structural equation modeling. Presenting the chosen cut-off values alongside the corresponding results facilitates the interpretation and evaluation of model quality directly within the results section. This approach aligns with reporting guidelines commonly adopted in studies using PLS, as suggested by Hair et al. [59]. We hope this explanation clarifies our rationale, and we remain open to further adjustments if needed.
OBSERVATION 9: I dont know what “source: own elaboration” means? Perhaps it is not necessary? Perhaps consider other factors to present in addition to SRMR, for example RMSEA and CFI, Line 459: I would delete the part of the pandemic.
RESPONSE 9: The phrase “source: own elaboration” has been removed from all sections of the manuscript where it previously appeared, as suggested. Regarding the first point, we acknowledge the importance of presenting additional model fit indices such as RMSEA and CFI. However, since the study was conducted using Partial Least Squares Structural Equation Modeling (PLS-SEM), these indices (RMSEA and CFI) are not typically reported, as they are primarily associated with covariance-based SEM. Instead, we have reported SRMR, which is one of the most appropriate global fit indices for PLS-SEM, in accordance with the guidelines provided by Hair et al. [59]. As recommended, we have removed the reference to the pandemic in line 459.
Reviewer 3 Report
Comments and Suggestions for Authors
The topic is one of interest and has potential, however, the current structure does not seem to lend itself to new or supportive inquiry. You mention that North American and Asian pet tourism is popular and want to bring in a Latin American perspective which would be great. The introduction, though, seems to have the same thoughts repeated throughout without truly adding the connection between cultures and the importance of including this in the literature. What else jumps out is that in the demographics, more than 72% of respondents are under the age of 34 and 96% are single, though that is not included in considerations for including pets. Both of those items seem highly significant in determining whether they are traveling with pets or not. You also mention cultural value, but do not provide any support or evidence of what you are referring to. The paper would serve value as you listed in the discussion section, thus being more specific and succinct in the introduction and throughout would build a better case.
Author Response
Dear Reviewer 3,
Thank you very much for your comments on our research paper. Our group has made the corrections you requested. Below are the comments and their responses:
OBSERVATION 1: The topic is one of interest and has potential, however, the current structure does not seem to lend itself to new or supportive inquiry. You mention that North American and Asian pet tourism is popular and want to bring in a Latin American perspective which would be great.
RESPONSE 1: We agree that highlighting the Latin American perspective more clearly strengthens the relevance of the study. In response to your suggestion, we have revised the introduction to better articulate the research gap by expanding on the limited empirical research on pet tourism in Latin America. Specifically, we replaced the original sentence with a more developed paragraph that emphasizes the lack of studies in this region despite the growing rates of pet ownership and evolving tourism behaviors in countries such as Chile. This addition reinforces the novelty and contribution of our work, positioning it as a response to the need for greater cultural and geographic diversity in pet tourism research. The revised paragraph has been incorporated into the introduction and is highlighted in blue for ease of review.
OBSERVATION 2: The introduction, though, seems to have the same thoughts repeated throughout without truly adding the connection between cultures and the importance of including this in the literature.
RESPONSE 2: In response to your comment, we carefully revised the introduction to eliminate redundancies and improve the logical structure and flow of the argument. In particular, we refined the narrative to avoid repeating ideas related to pet ownership trends and the growth of the pet tourism market. We also strengthened the cultural dimension of the study by explicitly addressing how the evolving role of pets—as integral members of the family—affects leisure practices, social identity, and travel-related consumption, especially in Latin American societies. This cultural framing is now clearly articulated as part of the research gap, reinforcing the importance of expanding the literature beyond the North American and Asian contexts, where most studies have been concentrated. These improvements help establish a clearer rationale for the study and its contribution to a more diverse and inclusive understanding of pet tourism. The revised content is included in the updated manuscript and has been highlighted in blue for ease of review.
OBSERVATION 3: What else jumps out is that in the demographics, more than 72% of respondents are under the age of 34 and 96% are single, though that is not included in considerations for including pets. Both of those items seem highly significant in determining whether they are traveling with pets or not
RESPONSE 3: We appreciate this thoughtful observation. Indeed, the demographic composition of our sample—particularly the high proportion of younger and single respondents—could play an important role in shaping pet-related travel behavior. While these factors were not explicitly modeled in the current analysis, we have now acknowledged their potential influence as a limitation in Section 6.1. Additionally, we have incorporated a suggestion in Section 6.3 (Future Research) to explore age and marital status as possible moderating variables in future studies, using techniques such as multi-group or interaction analysis.These changes have been added to the revised manuscript and are highlighted in blue.
OBSERVATION 4: You also mention cultural value, but do not provide any support or evidence of what you are referring to. The paper would serve value as you listed in the discussion section, thus being more specific and succinct in the introduction and throughout would build a better case.
RESPONSE 4: We agree that our previous mention of “cultural value” lacked sufficient clarity and supporting context. In response to your suggestion, we have revised the introduction to explicitly define what is meant by cultural value in the context of pet tourism. Specifically, we now highlight how the evolving role of pets as family members in Latin American societies influences social interactions and leisure-related consumer behavior. This clarification is supported with contextual data and placed directly after the discussion of the research gap in the introduction. The revised content is now included in the manuscript and has been highlighted in blue for your convenience. We appreciate your feedback, which has helped us enhance the conceptual precision and cultural framing of the study.
Round 2
Reviewer 1 Report
Comments and Suggestions for Authors
The authors have improved the manuscript substantially since the initial submittal (and were responsive to my comments and suggested revisions).
There are several inconsistencies in Table 3 in terms of factor names. "EAT" appears as a column header on the left side -- it seems that "EA" should appear instead. NOV and NO are both used in the table to denote Novelty. Both ATP and ATM are used for Attitudes. More broadly, there are inconsistencies in factor names across Figure 1, Table 2, Table 3, Table 4, and Table A1.
Notes at the bottom of Table 3: Fornell-Larcker statistics are on the diagonal and HTMT ratios appear above the diagonal.
Author Response
Dear Reviewer 1.
Thank you for pointing out the errors; we have corrected them. Please review the corrections (text highlighted in yellow and turquoise).
Reviewer 3 Report
Comments and Suggestions for Authors
Thank you for the changes. The paper feels more complete and comprehensive.
Author Response
Dear Reviewer 3,
Thank you for accepting our article. Your comments greatly helped us improve the presentation of our research.
Best regards.